# Municipal Wastewater Connection for Water Crisis and Jaundice Outbreaks in Shimla City: Present Findings and Future Solutions

**DOI:** 10.3390/ijerph191811266

**Published:** 2022-09-07

**Authors:** Ranju Kumari Rathour, Deepak Sakhuja, Arvind Kumar Bhatt, Ravi Kant Bhatia

**Affiliations:** Department of Biotechnology, Himachal Pradesh University Shimla, Shimla 171005, India

**Keywords:** Shimla, municipal wastewater, jaundice, pathogens, water crisis

## Abstract

The felicitous tourist destination “Hills Queen” and the capital city of Himachal Pradesh, an enticing state in the Himalayan region, are met with water crisis every year and jaundice outbreaks occasionally. In 2016, there was a severe jaundice outbreak in Shimla city. In a contemporaneous investigation, we attempted to trace out the possible reason for these crises in Shimla. Samples were collected month wise from different water-supply sources and their physicochemical and microbial loads were analyzed. The microbiological examination found a totally excessive microbial load (1.064 × 10^9^ cfu/mL on common) throughout the year with a maximum (>1.98 × 10^10^ cfu/mL) in the wet season and minimum (>3.00 × 10^7^ cfu/mL) in the winter. Biochemical and morphological evaluation confirmed that most of the water resources reported a high number of coliforms and Gram-negative microorganisms due to sewage-water infiltration. These microorganisms in the water are responsible for the liver infection that ultimately causes jaundice. For safe and potable water, infiltration of municipal wastewater must be prevented at any cost. Scientific disposal of wastewater and purification of uncooked water have to be conducted earlier than consumption or use for different domestic functions, to avoid water crises and fetal ailment outbreaks in the near future.

## 1. Introduction

Water is the primary component of every living organism, with fluids derived from numerous sources on Earth. It is necessary for all known forms of life on Earth and covers 71% of the planet’s surface [1,2,3]. India is endowed with a vast range of snow-capped Himalayas and a network of rivers that contribute to the country’s water supply. Rivers play a vital role in the lives of the Indian people by providing water for drinking and various other anthropogenic activities [4,5]. There is a definite link between water availability and the prevention of numerous water-borne illnesses. Although access to clean drinking water has increased in nearly every corner of the world over the last several decades, however one billion people (globally) still lack access to drinking and sanitation [6]. The infiltration of home and industrial effluent into the water sources has degraded the available pure water. Drinking water polluted by various chemical or physical sources has the biggest impact on human health, particularly in underdeveloped nations [7]. However, it is anticipated that by 2025, more than half of the world’s population will be vulnerable to water-related hazards.

According to one of WHO’s reports, by 2030, water shortage in some developing countries may surpass 50%, resulting in 80% of water-borne or water-linked human diseases owing to biological pollution of drinking water [8]. Drinking water should ideally not include any harmful microorganisms or bacteria suggestive of fecal contaminations (increased amount of *E. coli* and other coliforms) [9,10]. Many cities and other rural/urban regions have had outbreaks of water-borne infections as a result of inadequate management of water supplies and poor waste disposal. Consumption of drinkable water with sewage water in the Municipal Corporation of Shimla in 2016 resulted in a jaundice epidemic, which infected more than 1600 people and reported 16 casualties [11].

Authorities halted the supply of drinking water from polluted sources (the city’s primary water sources), exacerbating Shimla’s water crisis. The epidemic was caused by a rise in fecal coliform concentrations in drinking water. Aside from jaundice, other disorders, such as ear infections, dysentery, typhoid fever, viral and bacterial gastroenteritis, and so on, are also associated with the presence of fecal coliforms such as *Enterobacter*, *Klebsiella*, *Citrobacter*, *E.coli*, etc. [12]. Regular extensive treatment and analysis of drinking water is required to keep an eye on and to minimize the microbial burden. Controlling microbiological contamination at all checkpoints might be a solution to water-borne illnesses and its link to water crises in all mega cities and significant tourist locations, such as Shimla, to prevent against any future epidemic disasters and water crises.

## 2. Materials and Methods

### 2.1. Area under Investigation

Shimla, a picturesque hill station and popular tourist destination in the Himalayas (Figure 1) faced water crisis in 2016. The city is spread out across an uneven altitude of 2100 m above sea level. This magnificent hill station is located between 31.06° N and 77.13° E. The city has a total size of 25 km^2^ [13,14]. Shimla’s water-supply system dates back to 1875, and additional water sources have been located to supplement the supply in order to fulfill the city’s ever-increasing demand for water. Dhalli Catchment area, Cherot Nallah, Jagroti Nallah, Chair Nallah, Gumma Khad, and Ashwani Khad pump and provide surface water from rivulets at various altitudes. The total installed capacity is 47.54 million liters. Four water treatment plants exist in Gumma Khad, Ashwani Khad, Cherot Nallah and Dhalli Catchment area.

### 2.2. Sample Collection

Water samples were collected as per WHO guidelines from different water-supply sources every month of one year in sterile bottles and brought to laboratory for further processing. 

### 2.3. Analysis of Physiochemical Parameters

Various physio-chemical parameters such as temperature, pH, TDS, salt concentration, color and conductivity were recorded. 

### 2.4. Analysis of Microbiological Load

The microbial load was analyzed by standard plate-count technique using a serial dilution agar-plate method up to 10^−6^-fold dilution of samples. Further, 100 µL of each dilution was spread on different basic as well as selective media (Nutrient, MacConkey, EMB, Endo) plates and incubated at 37 °C for 24 h [4]. After 24 h, the plates were analyzed for different morphological appearances on different media and then further sub-cultured to obtain pure culture by repeated streaking.

### 2.5. Identification of Microbes

Selected isolates were examined for various morphological as well as biochemical features such as colony morphology, Gram’s staining, methyl red, oxidase test, catalase test, glucose, lactose, maltose, sorbitol fermentation test, Vogues–Proskaure test, hydrolysis of casein, etc., following the standard protocols.

### 2.6. DO, BOD and COD of Collected Samples

Standard protocols were used for analysis of dissolved, biological and chemical oxygen demands [14,15].

## 3. Results

### 3.1. Collection of Samples

Samples were chronologically collected from source, i.e., raw water, to finale filtration tanks and finally to distribution points, as shown in Figure 2 of the three main water-supply sources. 

### 3.2. Analysis of Physicochemical Parameters

The physical and chemical parameters of water were analyzed at the time of sampling and data generated are shown in Table 1, Table 2 and Table 3. All the water sources, i.e., Dhalli Catchment area, Cherot Nallah/Jagroti Nallah and Ashwani Khad, showed average pH (8.5), TDS (108 ppm), conductivity (131 µS) and salt (70 ppm), which were under the permitted limit of WHO norms for drinking water [10,16,17]. The above-mentioned range of parameters might be due to the presence of rocks and alkaline soil in the catchment area of water sources, but these are within the permitted limits and water is safe to use for domestic purposes. Average water temperature of all the sources was found in the range of 15–25 °C and it provides an ambient temperature for the survival of mesophilic microorganisms. This low range of temperature is due to the fact that all these water sources are fed by melting snow caps and dense forest streams. However, all these parameters are within the permitted range of WHO guidelines (Table 4), but at the same time, these factors act as add-on factors during infiltration of litter and sewage to the water sources and help in the proliferation of opportunistic pathogens.

### 3.3. Analysis of Microbiological Load

The standard agar-plate method was used to analyze the microbial load in the water samples. *E*. *coli* and *Klebsiella* sp. were most-commonly present lactose-fermenting *Enterobacter*, while *Pseudomonas* sp. and *Proteus* sp. were non-lactose fermenting isolates commonly present in all samples (Figure 3). More than 8.97 × 10^9^ cfu/mL of bacteria were observed in different samples based of their morphology on different media and biochemical analysis. Svanevik and Lunestad [18] also used this method to study microbial contaminations in water. During the microbial load analysis, it was found that all the water samples were heavily contaminated with microorganisms, especially opportunistic pathogens, i.e., *E. coli* and *Pseudomonas* sp., which might be responsible for the jaundice, and this microbial load was very high as compared to that of the standard limits of WHO.

Further, it has been also found that microbiological load was >2.25 × 10^7^ cfu/mL in January–February, >4.36 × 10^9^ cfu/mL March–April, 4.72 × 10^9^ cfu/mL May–June, >4.064 × 10^10^ cfu/mL July–August, September–October 5.54 × 10^9^ cfu/mL and >4.01 × 10^9^ cfu/mL in November–December. Microbiological contamination was almost similar in all the samples. *E. coli* and *Pseudomonas* sp. were prominently observed throughout the year. Presence of *E. coli* and *Pseudomonas* sp. was also even recorded in tap water. *Klebsiella* sp. and *Proteus* sp. were also reported in the Dhalli water treatment plant during rainy session, which might be due to infiltration of sewage from nearby villages/urban localities to the water streams. The standard limit is 2.7 × 10^2^ cfu/mL [19], while the microbiological load was found to be very high as compared to standard limits. Month-wise microbiological load of all the collected samples is shown in Table 5. 

### 3.4. DO, BOD and COD Analysis

DO, BOD and COD are measured by the amount of organic compounds in water. Biochemical/ biological oxygen demand is the amount of dissolved oxygen needed by a biological system in water to break down organic materials present in a given water sample at certain temperature over a specific time period. It is most commonly expressed in milligrams of oxygen consumed per liter of sample during 5 days of incubation at 20 °C and used as an index to determine organic pollution. The maximum BOD was observed in raw water of Churat Nallah, i.e., 16.8 ± 0.053 mg/L, followed by the raw water of Ashwini Khad water treatment plant (14.4 ± 0.065 mg/L). BOD value of public-tap water was in range from 0.2 to 1.0 mg/L. BOD were calculated on the basis of the difference between the DO value of the water sample on first ady and the DO value after a specific time interval (5 days).

Chemical oxygen demand (COD) is an indicative measure of the amount of oxygen that can be consumed by chemical reactions in the solution under test. It is commonly expressed in mass of oxygen consumed over volume of solution, which in SI units is milligrams per liter (mg/L). Maximum COD was observed with the raw water of Churat Nallah (0.384 ± 0.005 mg/L). Kadam and Agrawal [20] also analyzed the BOD and COD values to ensure potability of water. The DO, BOD and COD of samples are shown in Table 6. BOD and COD are measures of water contamination. The higher value of BOD indicates the presence of more biological contaminations in water [21]. The total bacterial count and the coliform density were directly related to the biochemical oxygen demand, but inversely related to the dissolved oxygen. Thus, tests for dissolved oxygen are of the utmost importance for controlling water pollution [22].

## 4. Discussion

Different water samples were collected month-wise from different sites of the main water-supply sources of Shimla and almost all physiochemical parameters were found to be suitable for potable water according to WHO standards, except microbial load. Kistemann et al. [23] also analyzed physiochemical parameters to ensure potability of drinking water. Behailu et al. [24] also analyzes the potability of drinking water and found that the average pH of all samples was in the range 7.6–8.2. The average pH and temperature of all samples were found to be 8.5 and 19 °C, respectively. The pH and temperature of all water pumping sources were ambient and conducive for the growth of opportunistic pathogens such as *E. coli, Pseudomonas* sp. *Klebsiella* sp. and *Proteus* sp. etc. and resulted in higher microbial growth, which supports our findings that a microbiological load higher than that of the permissible range of water potability standards might be a possible reason for jaundice outbreak in Shimla. Further, shortage in the rainfall intensified the issue and increases the concentration of microbes in the water sources, resulting in water shortage and crisis in the hill region. The observed TDS in the present study was varied from 75 to 225 ppm. According to the WHO, the TDS standard for drinking water is <1000 ppm. The collective amount of all dissolved cation and anions in the water is known as TDS and it is mainly associated with conductivity [25]. High TDS increases the density of water, decreases solubility of gases such as oxygen, and ultimately makes the water unsuitable for drinking [26]. The measure of competence of water to pass electrical flow is its conductivity, which is directly related to the concentration of ions in water coming from dissolved salts and inorganic materials such as alkalis, chlorides, sulfides and carbonate compounds. Conductivity of drinking water is in the range of 5–50 mS/m, (5,000,000 μS/m) [27]. Conductivity was found to be high in the month of July to September when microbial load was high. Increase in conductivity leads to increase in microbial load as minerals and matter provide necessary metabolites and factors for the rapid growth of microorganisms [28]. The lower TDS and conductivity ensure the potability of water. 

BOD and COD of water samples were also analyzed using standard protocol and were found to be higher than the permissible limit. *E. coli* and *Pseudomonas* sp. were prominently observed in all the samples throughout the year. Both of these bacteria are opportunistic pathogens for humans. *E. coli* is an indicator organism of water pollution. Water was highly contaminated with *Klebsiella* sp. in the months of June, July, August and September. Onyango et al. [29] also reported presence of *E. coli* and other coliforms in drinking water during their study. Presence of *E. coli* or fecal contaminants in water is the main cause for the outbreak of jaundice [30,31]. According to a report [32], the DO of drinking water must be in the range 2–5 mg/L, BOD of drinking water 2–10 mg/L and COD 5–10 mg/L. According to a report [33], the jaundice epidemic in 2016 was caused by contaminated water from Ashwami Khad, from where Shimla used to acquire its largest share of water. After the report of jaundice cases, government officials stopped the water supply from these contaminated water sources, which increased the water shortage in the city and ultimately deepened the water crisis. 

## 5. Conclusions and Future Perspectives

All the water distribution channels in Shimla were badly contaminated with various opportunistic pathogens, as revealed through our investigation. Fortunately, much less contamination appeared in the distribution line after treatment, but microbial load was still beyond the permitted limit. Results of this research could help the MC/IPH officials to take remediate measures in good time. Authorities should also look for alternative water sources to prevent any water crisis in the future that may also arise due to a shortage of rainfall in the catchment areas. The efforts will also help to establish a cleaning schedule for the MC water-supply sources and create proper wastewater treatment facilities for the villages/towns situated nearby the catchment areas, to prevent the infiltration of domestic and industrial sewage, which will help to provide safe drinking water to the consumers in MC Shimla. Moreover, based upon our findings, the MC/IPH department may install a heavy-duty UV system to control the heavy microbial load. Using these recommended corrective measures, such future disasters could be avoided, and the public of such areas will have potable drinking water free from all types of contaminants and opportunistic pathogens. 

## Figures and Tables

**Figure 1 ijerph-19-11266-f001:**
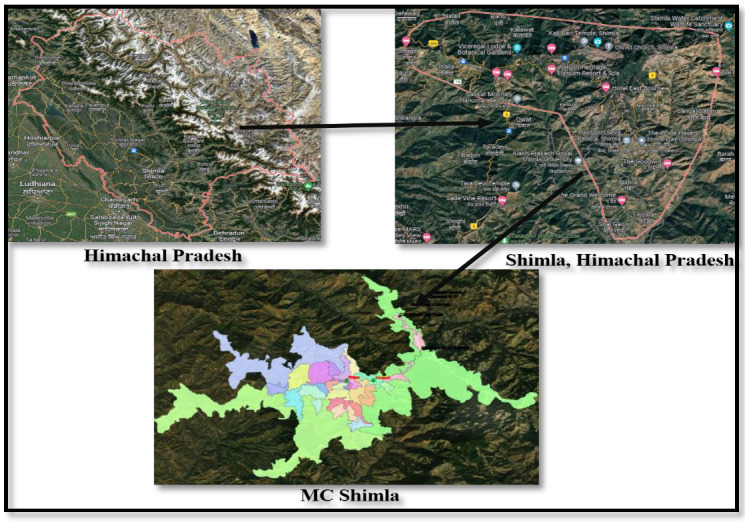
Geographical location of study area.

**Figure 2 ijerph-19-11266-f002:**
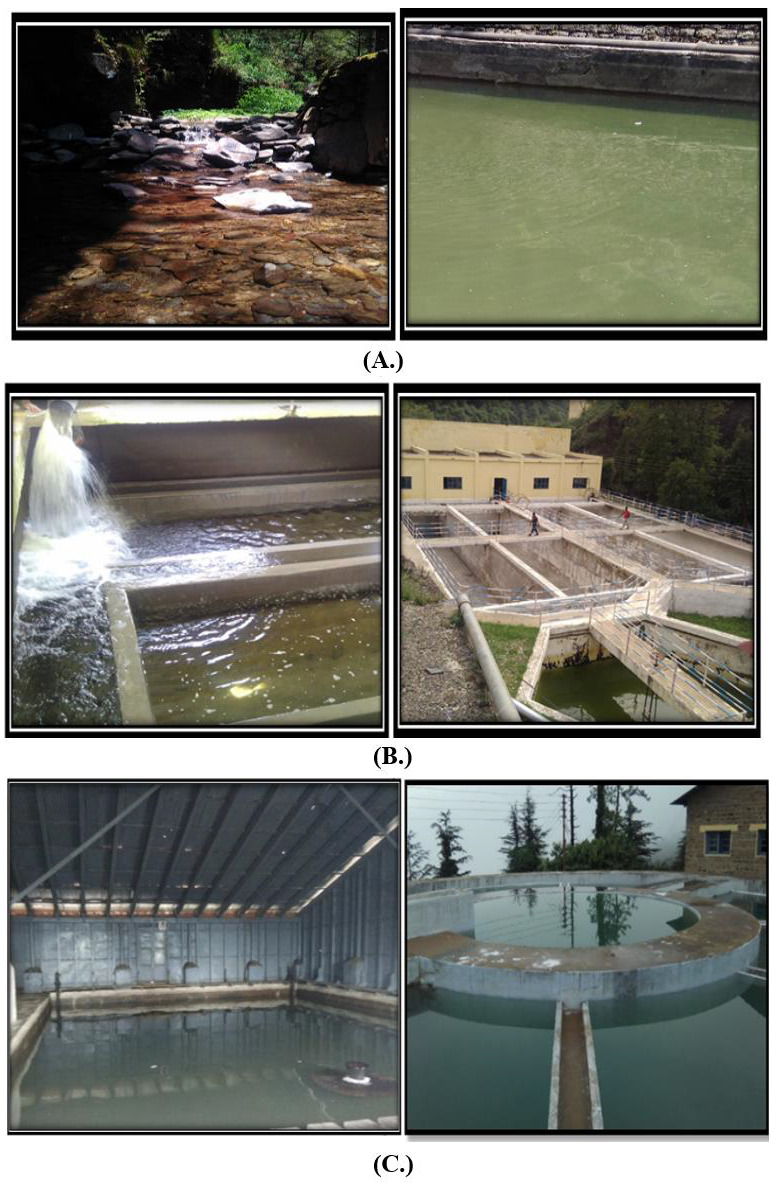
Different sampling sites investigated during course of study. (**A**) Churat Nallah, (**B**) Ashwani Khad hard workers (**C**) Water treatment plant Dhalli.

**Figure 3 ijerph-19-11266-f003:**
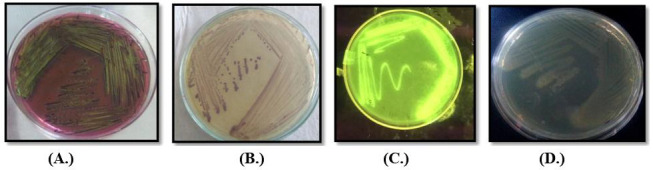
Colony morphology of different microbes on different media (**A**.) *E. coli* on EMB (**B**.) *Klebsiella* on *Klebsiella* selective agar and (**C**.) *Pseudomonas* sp. on Kings B medium under UV transilluminator and (**D**.) *Proteus* sp. on nutrient agar.

**Table 1 ijerph-19-11266-t001:** Analysis of samples collected from source Churat/Jagroti Nallah.

Parameters	WHO Standards	Raw Water Churat Nallah						Raw Water Jagroti Nallah					
		Jan–Feb	March–April	May–June	July–August	September–October	November–December	January–February	March–April	May–June	July–August	September–October	November–December
**Color**	Colorless	Colorless	Colorless	Colorless	Slightly Turbid	Colorless	Colorless	Colorless	Colorless	Colorless	Slightly turbid	Colorless	Colorless
**pH**	6.5–8.5	8.4 ± 0.28	8.2 ± 0.27	8.5 ± 0.15	8.2 ± 0.11	8.9 ± 0.44	8.2 ± 0.33	8.6 ± 0.27	8.4 ± 0.56	8.4 ± 0.20	8.0 ± 0.15	8.5 ± 0.57	8.2 ± 0.44
**Temp (°C)**		15.7 ± 0.56	17.3 ± 0.33	18.2 ± 0.11	16.2 ± 0.25	18.7 ± 0.57	12.7 ± 0.24	16.2 ± 0.20	16.8 ± 0.33	24.7 ± 0.57	25.1 ± 0.44	19.2 ± 0.56	20.1 ± 0.11
**TDS (ppm)**	300	86.9 ± 0.25	93.5 ± 0.11	105.0 ± 0.27	71.8 ± 0.24	103.0 ± 0.28	132.9 ± 0.65	89.5 ± 0.19	95.3 ± 0.25	131.0 ± 0.28	120.0 ± 0.33	101.0 ± 0.27	72.3 ± 0.19
**Conductivity (µS/cm)**	250	123.5 ± 0.19	107.5 ± 0.19	148.8 ± 0.25	100.2 ± 0.20	141.9 ± 0.19	98.0 ± 0.25	113.5 ± 0.57	117.2 ± 0.28	141.5 ± 0.24	136.2 ± 0.56	181.3 ± 0.28	101.8 ± 0.33
**Salt conc. (mg/L)**	30–60	59.8 ± 0.15	76.3 ± 0.25	69.4 ± 0.56	48.9 ± 0.33	54.2 ± 0.11	67.5 ± 0.56	63.8 ± 0.44	83.3 ± 0.19	69.4 ± 0.60	55.6 ± 0.25	87.3 ± 0.20	50.6 ± 0.44

**Table 2 ijerph-19-11266-t002:** Analysis of samples collected from source Ashwani Khad.

Parameters	Raw Water						Filtration Tank					
	Jan–Feb	March–April	May–June	July–August	September–October	November–December	January–February	March–April	May–June	July–August	September–October	November–December
**Color**	Colorless	Colorless	Colorless	Turbid	Colorless	Colorless	Colorless	Colorless	Colorless	Turbid	Colorless	Colorless
**pH**	8.4 ± 0.25	8.2 ± 0.15	8.6 ± 0.29	8.3 ± 0.23	8.5 ± 0.56	8.3 ± 0.28	8.3 ± 0.24	8.4 ± 0.27	8.5 ± 0.22	8.2 ± 0.17	8.2 ± 0.24	8.6 ± 0.29
**Temp (°C)**	15.6 ± 0.33	17.2 ± 0.27	21.4 ± 0.43	22.0 ± 0.53	18.5 ± 0.25	16.7 ± 0.20	15.8 ± 0.44	16.4 ± 0.53	24.5 ± 0.53	20.1 ± 0.27	19.2 ± 0.25	15.8 ± 0.48
**TDS (ppm)**	118.0 ± 0.43	116.5 ± 0.53	128.0 ± 0.24	123.0 ± 0.56	125.0 ± 0.22	137.0 ± 0.25	123.0 ± 0.27	123.0 ± 0.25	126.0 ± 0.11	101.8 ± 0.20	110.0 ± 0.29	123.4 ± 0.27
**Conductivity (µS/m)**	122.3 ± 0.13	128.3 ± 0.33	180.3 ± 0.13	86.6 ± 0.19	98.3 ± 0.45	107.0 ± 0.26	100.8 ± 0.32	118.3 ± 0.26	190.0 ± 0.25	72.3 ± 0.29	96.2 ± 0.19	127 ± 0.57
**Salt conc. (ppm)**	85.0 ± 0.34	83.0 ± 0.44	87.0 ± 0.23	60.7 ± 0.26	74.2 ± 0.34	82.0 ± 0.29	88.8 ± 0.45	83.7 ± 0.24	91.9 ± 0.18	50.6 ± 0.25	83.4 ± 0.54	89.0 ± 0.20

**Table 3 ijerph-19-11266-t003:** Analysis of samples collected from source Dhalli WTP.

Parameters	January–February			March–April			May–June			
	Raw Water	Filtration Tank	Storage Tank	Raw Water	Filtration Tank	Storage Tank	Raw Water	Settling Tank	Public Tap 1	Public Tap 2
**Color**	Colorless	Colorless	Colorless	Colorless	Colorless	Colorless	Colorless	Colorless	Colorless	Colorless
**pH**	8.3 ± 0.19	8.5 ± 0.26	8.4 ± 0.28	8.7 ± 0.25	8.5 ± 0.27	8.8 ± 0.28	8.30 ± 0.25	8.45 ± 0.19	8.67 ± 0.29	8.54 ± 0.23
**Temp. (°C)**	12.7 ± 0.28	12.4 ± 0.27	13.0 ± 0.25	16.3 ± 0.57	17.5 ± 0.28	20.5 ± 2	15.7 ± 0.19	16.4 ± 0.32	17.3 ± 0.57	16.8 ± 0.28
**TDS (ppm)**	132.2 ± 0.34	118.5 ± 0.54	129 ± 0.23	125 ± 0.43	145.3 ± 0.43	106.8 ± 0.19	102.2 ± 0.28	118.8 ± 0.23	145 ± 0.54	149.8 ± 0.35
**Conductivity (µS/m)**	74.3 ± 0.45	108.6 ± 0.32	115 ± 0.36	114 ± 0.28	127.6 ± 0.34	148.3 ± 0.32	84.8 ± 0.21	94.5 ± 0.36	102 ± 0.43	97.6 ± 0.38
**Salt conc. (ppm)**	85.3 ± 0.21	78.1 ± 0.17	94.2 ± 0.43	88.3 ± 0.36	78.2 ± 0.29	74.5 ± 0.44	87.4 ± 0.57	78.6 ± 0.27	78.5 ± 0.29	75.8 ± 0.27
**Parameters**	**July–August**			**September–October**			**November–December**			
	**Raw Water**	**Settling Tank**	**Storage Tank**	**Filtrated Water**	**Public Tap 1**	**Public Tap 2**	**Raw Water**	**Settling Tank**	**Public Tap 1**	**Public Tap 2**
**Color**	Colorless	Turbid	Slightly turbid	Slightly turbid	Colorless	Colorless	Colorless	Colorless	Colorless	Colorless
**pH**	8.9 ± 0.21	8.7 ± 0.36	8.9 ± 0.11	9.7 ± 0.52	8.7 ± 0.38	8.6 ± 0.17	8.23 ± 0.33	8.15 ± 0.23	8.79 ± 0.20	8.33 ± 0.19
**Temp. (°C)**	18.0 ± 2	22.8 ± 2	18.7 ± 2	18.0 ± 2	20.3 ± 2	24.0 ± 2	12.70 ± 2	15.4 ± 2	16.3 ± 2	16.80 ± 0.20
**Cond (µS/m)**	89.9 ± 0.18	83.5 ± 0.19	103 ± 0.43	145 ± 0.16	107 ± 0.17	94.6 ± 0.28	132.9 ± 0.42	118.8 ± 0.37	135 ± 0.42	139.8 ± 0.23
**TDS (ppm)**	115.0 ± 0.54	122.8 ± 0.39	141.9 ± 0.15	212.0 ± 0.37	150.0 ± 0.26	132.8 ± 0.27	98 ± 0.31	100.8 ± 0.18	107 ± 0.28	97.6 ± 0.29
**Salt Conc. (ppm)**	58.8 ± 0.24	61.7 ± 0.33	69.5 ± 0.26	98.0 ± 0.53	72.8 ± 0.39	66.1 ± 0.36	67.5 ± 0.28	68.5 ± 0.43	72.8 ± 0.43	65.1 ± 0.54

**Table 4 ijerph-19-11266-t004:** Standard limits of different parameters in potable water.

Sr. No.	Parameter	Permissible Limit	Instrument Used for Analysis
1	**Color**	Colorless	-
2	**pH**	6.5–8.5	pH meter
3	**Temp. (°C)**	Varies with environmental conditions	Thermometer
4	**TDS (ppm)**	1000	TDS meter
5	**Conductivity (µS/m)**	400	Conductivity meter
6	**Salt conc. (ppm)**	250	Conductivity meter
7	**BOD (ppm)**	1–2	Titration
8	**COD (ppm)**	10	Titration

**Table 5 ijerph-19-11266-t005:** Microbial load of different water samples.

Months	Microbial Load									
	Jagroti Nallah	Churat Nallah	Ashwani Khad		Dhalli WTP					
	Raw Water	Raw Water	Raw Water	Filtration Tank	Raw Water	Settling Tank	Storage Tank	Filtration Tank	Public Tap 1	Public Tap 2
**January–February**	1.81 × 10^6^	2.31 × 10^6^	1.82 × 10^7^	1.82 × 10^5^	1.93 × 10^3^	2.43 × 10^3^	4.76 × 10^4^	1.20 × 10^4^	3.26 × 10^2^	3.54 × 10^2^
**March–April**	1.31 × 10^6^	1.82 × 10^7^	2.02 × 10^7^	2.02 × 10^7^	5.42 × 10^2^	3.12 × 10^3^	5.43 × 10^3^	3.80 × 10^7^	6.80 × 10^2^	6.38 × 10^2^
**May–June**	2.11 × 10^7^	5.43 × 10^7^	2.82 × 10^7^	2.13 × 10^7^	5.76 × 10^4^	5.42 × 10^4^	4.80 × 10^7^	3.76 × 10^6^	5.93 × 10^3^	4.43 × 10^2^
**July–August**	8.42 × 10^7^	1.18 × 10^7^	6.82 × 10^7^	5.80 × 10^7^	3.80 × 10^7^	6.43 × 10^5^	4.93 × 10^7^	3.82 × 10^7^	8.82 × 10^2^	5.42 × 10^3^
**September–October**	1.98 × 10^7^	6.21 × 10^7^	4.32 × 10^7^	3.76 × 10^6^	6.20 × 10^6^	6.20 × 10^6^	8.82 × 10^7^	4.20 × 10^6^	2.93 × 10^3^	1.13 × 10^3^
**November–December**	4.52 × 10^6^	3.11 × 0^3^	2.75 × 10^7^	2.76 × 10^6^	4.52 × 10^7^	6.12 × 10^3^	6.20 × 10^6^	4.27 × 10^6^	3.43 × 10^2^	2.42 × 10^2^

**Table 6 ijerph-19-11266-t006:** BOD and COD for different water samples.

Sr. No	Sample	DO (0 Day) mg/L	DO (5 Days) mg/L	BOD (mg/L)	COD (mg/L)
01.	Blank	8 ± 0.078	7.2 ± 0.054	4.8 ± 0.049	07.00 ± 0.022
02.	Ashwani Khad raw water	4.8 ± 0.065	2.4 ± 0.022	14.4 ± 0.065	0.0228 ± 0.005
03.	Ashwani Khad Filtration tank	4.6 ± 0.028	2.4 ± 0.029	13.2 ± 0.080	0.0126 ± 0.002
04.	Jagroti Nallah raw water	4.8 ± 0.022	3.9 ± 0.077	5.4 ± 0.76	0.0128 ± 0.001
05.	Churat Nallah raw water	3.6 ± 0.053	0.8 ± 0.032	16.8 ± 0.053	0.384 ± 0.005
06.	Churat Nallah filtration tank	3.6 ± 0.049	2 ± 0.078	9.6 ± 0.069	0.348 ± 0.005
07.	Dhali WTP Sand filtration	3.2 ± 0.054	1.6 ± 0.029	9.6 ± 0.062	0.0164 ± 0.002
08.	Dhali WTP Settling tank	3.4 ± 0.065	1.60 ± 0.056	10.8 ± 0.045	0.0125 ± 0.003
09.	Dhali WTP filtration tank	3 ± 0.052	1.2 ± 0.055	10.8 ± 0.055	0.0064 ± 0.002
10.	Blank	2 ± 0.039	1.6 ± 0.062	0.4 ± 0.059	06.80 ± 0.065
11.	Churat Nallah Raw water	1.2 ± 0.054	0.5 ± 0.067	0.7 ± 0.063	0.0232 ± 0.003
12.	Dahli WTP	0.9 ± 0.022	0.3 ± 0.060	0.6 ± 0.068	0.0141 ± 0.001
13.	Chlorinated water	0.8 ± 0.065	0.5 ± 0.074	0.3 ± 0.065	0.0122 ± 0.003
14.	Public tap 1 Dhalli	1.6 ± 0.067	1 ± 0.075	0.6 ± 0.047	0.254 ± 0.007
15.	Public tap 2 Dhalli	1.2 ± 0.056	0.9 ± 0.025	0.3 ± 0.071	0.282 ± 0.006
16.	Public tap 1 Sanjauli	1.0 ± 0.046	0.9 ± 0.072	0.3 ± 0.067	0.276 ± 0.007
17.	Public tap 2 Sanjauli	1.3 ± 0.022	1.0 ± 0.069	0.2 ± 0.054	0.287 ± 0.003
18.	Public tap 1 Summer hill	1.3 ± 0.049	0.8 ± 0.058	0.5 ± 0.041	0.256 ± 0.004
19.	Public tap 2 Summer hill	1.4 ± 0.054	0.9 ± 0.055	0.4 ± 0.065	0.285 ± 0.006
20.	Public tap 3 Summer hill	1.2 ± 0.076	1.0 ± 0.065	0.6 ± 0.065	0.263 ± 0.005

## Data Availability

Not applicable.

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
