# Peer review of "Municipal Wastewater Connection for Water Crisis and Jaundice Outbreaks in Shimla City: Present Findings and Future Solutions"

_ijerph, 2022, doi:10.3390/ijerph191811266_

Round 1
Reviewer 1 Report (Previous Reviewer 1)
The correction of the comments made to your work is appreciated, there is not much news but it is a work that gives good results and that are necessary to find possible solutions to your pollution problems.
Author Response
Query 1: The correction of the comments made to your work is appreciated, there is not much news but it is a work that gives good results and that are necessary to find possible solutions to your pollution problems.
Response: Thank you for your valuable concerns. We have improvised the manuscript according as per the suggestions.
Reviewer 2 Report (Previous Reviewer 2)
The authors should be attention to the following comment.
· The manuscript is the same as a report. It must be improved.
· In this manuscript, the statistical relationship between the prevalence of hepatitis or the occurrence of jaundice and the microbial load of drinking water has been NOT established. According to the title of the manuscript, it is the goal of the work. Please justify.
· In this research, the authors conclude that opportunistic pathogens i.e. E. coli and Pseudomonas sp. may act as hosts for the Hepatitis virus. This result is speculative, and it needs to approve by a confidential Hepatitis virus analysis method.
· In section 2.3. Analysis of physicochemical parameters, and 2.6. DO, BOD, and COD of collected samples, tools, and equipment water analysis must be presented. Please done.
· Chemical oxygen demand is not a proper parameter for water treatment evaluation. In this condition TOC is essential. Please clarify.
Author Response
Query 1: In this manuscript, the statistical relationship between the prevalence of hepatitis or the occurrence of jaundice and the microbial load of drinking water has been NOT established. According to the title of the manuscript, it is the goal of the work. Please justify.
Response: Thank you for your comment. The Jaundice outbreak in Shimla (2016) was reported to be because of increased concentration of fecal coliforms in drinking water (Bodh, 2016). Although there was not specific reports regarding direct relationship between microbial load and hepatitis. But it was observed that bacterial infections are associated with a high level of HBV replication (Li et al. 2015). In present study author also try to find out possible reasons of the Jaundice outbreak. The higher microbial contaminations was observed in water which can act as a possible source for replication of virus and justifies the finds of Li and co-workers.
Query 2: In this research, the authors conclude that opportunistic pathogens i.e. E. coli and Pseudomonas sp. may act as hosts for the Hepatitis virus. This result is speculative, and it needs to approve by a confidential Hepatitis virus analysis method.
Response: Thank you for your comment. These results were hypothetically confirmed with the find of Li et al. (2015) who, in a study reported that higher the level of microbes, higher the replication of virus and higher will the chances of jaundice. Author highly consider reviewers comment and will work on it for further investigations.
Query 3: In section 2.3. Analysis of physicochemical parameters, and 2.6. DO, BOD, and COD of collected samples, tools, and equipment water analysis must be presented. Please done.
Response: Thank you for your comment. Needful has been done. The equipment used for analyses has also been mentioned (Table 4).
Query 4: Chemical oxygen demand is not a proper parameter for water treatment evaluation. In this condition TOC is essential. Please clarify.
Response: Thank you for your comment. COD is a measure of organic matter and oxygen demand; however, the method of oxygen depletion is through chemical oxidation reactions as opposed to biological activity. TOC and DOC provide a measure of the total amount of carbon in a sample through determination of CO2 generation from an oxidation reaction. COD is easy to perform with the right equipment (COD analyser and digestor) and can be done in 2 hours. BOD usually takes 5 days and TOC used to require large sophisticated pieces of equipment that could measure the sample in minutes, but was cost prohibitive.
Reviewer 3 Report (Previous Reviewer 3)
The manuscript reports on the quality of water in Shimla. Water samples were collected once a month through out one year to see the variability in chemical and biological parameters of water. The context of the study is a jaundice outbreak in 2016 which was connected with water crisis and a heavy water contamination. This report is of importance for the local community and may serve as an example for other areas where clean water supply is limited.
Please add the information on the year in which the study was conducted.
Authors found that water was particularly contaminated in June-August period. Do Authors have any explanation to this finding?
Please change the numbering of Tables. At present Table 1 is divided into Table 1a, 1b, 1c - these may go with separate numbers.
Author Response
Query 1: The manuscript reports on the quality of water in Shimla. Water samples were collected once a month through-out one year to see the variability in chemical and biological parameters of water. The context of the study is a jaundice outbreak in 2016 which was connected with water crisis and a heavy water contamination. This report is of importance for the local community and may serve as an example for other areas where clean water supply is limited.
Response: Thank you for your comment. We have improvised the manuscript according as per the suggestions.
Query 2: Please add the information on the year in which the study was conducted.
Response: Thank you for your comment. Needful as suggested has been done.
Query 3: Authors found that water was particularly contaminated in June-August period. Do Authors have any explanation to this finding?
Response: Thank you for your comment. The weather of Shimla in July and august remains quite pleasant and comfortable. The temperature of Shimla in these months is generally in range 21-28°C. Shimla remains incredibly wet during july and august and it rains almost every single day. The water parameter are different might be due to heavy rainfall, higher moisture content and increased amount of water in natural resources, and excess amount of soil and sandy substances in natural resources.
Query 4: Please change the numbering of Tables. At present Table 1 is divided into Table 1a, 1b, 1c - these may go with separate numbers.
Response: Thank you for your comment. Needful as suggested has been done.
References
- Li, W., Jin, R., Chen, P., Zhao, G., Li, N., & Wu, H. (2015). Clinical correlation between HBV infection and concomitant bacterial infections. Scientific Reports, 5(1), 1-6.
- Bodh A., (2016). Jaundice outbreak reaching epidemic proportions in Shimla. A report, The Times of India. https://timesofindia.indiatimes.com/city/shimla/Jaundice-outbreak-reaching-epidemic-proportions-in-Shimla/articleshow/50517708.cms
This manuscript is a resubmission of an earlier submission. The following is a list of the peer review reports and author responses from that submission.
Round 1
Reviewer 2 Report
The authors should be attention to the following comment.
· The manuscript is the same as a report. It must be improved.
· In this manuscript, the statistical relationship between the prevalence of hepatitis or the occurrence of jaundice and the microbial load of drinking water has been NOT established. According to the title of the manuscript, it is the goal of the work. Please justify.
· In this research, the authors conclude that opportunistic pathogens i.e. E. coli and Pseudomonas sp. may act as hosts for the Hepatitis virus. This result is speculative, and it needs to approve by a confidential Hepatitis virus analysis method.
· Line 2, wastewater, please write all of the waste water terms in the form of wastewater.
· Line 2, Jaundice or hepatitis outbreak, I think this is a mistake. I proposed the authors revised the whole of the manuscript.
· Line 10 to 12, The microbiological examination found a totally excessive microbial load (1.064x109 cfu/ml on common) throughout the year with maximum (>1.98x1010 cfu/ml) in wet sessions and minimum (>3.00x107 cfu/ml) in winter. Are you sure? Maximum in wet sessions and minimum in winters? I think the sentence needs to revise, please do.
· Figure 1, Geographical location of the study area is unclear. Please improve the figure.
· In section 2.3. Analysis of physicochemical parameters, and 2.6. DO, BOD, and COD of collected samples, tools, and equipment water analysis must be presented. Please done.
· Chemical oxygen demand is not a proper parameter for water treatment evaluation. In this condition TOC is essential. Please clarify.
· Unfortunately, Table 1 is very messy. Please correct it.
Reviewer 3 Report
The article describes a systematic potable water quality studies in Shimla, India. The physic-chemical parameters of water from different sources are examined followed by the microbial study. Data support the reasons for jaundice outbreak in 2016 in the area. Authors propose way how to ameliorate water quality. The article is relevant for the readers of the Int. J. Environ. Res. Public Health. I recommend publishing it with minor changes.
The used abbreviations would be explained on the first appearance in the text.
A table summarising the acceptable range of physicochemical parameters of potable water would add to the clarity of the manuscript.
A short explanation, based probably on the typical weather, why water parameters are slightly different in July-August than in other months would be welcome (for those not knowing the climate particularities of Shimla).